# Rab GTPases: Switching to Human Diseases

**DOI:** 10.3390/cells8080909

**Published:** 2019-08-16

**Authors:** Noemi Antonella Guadagno, Cinzia Progida

**Affiliations:** Department of Biosciences, University of Oslo, 0316 Oslo, Norway

**Keywords:** GTPases, Rab proteins, membrane trafficking, neurodegeneration, cancer, intracellular pathogens

## Abstract

Rab proteins compose the largest family of small GTPases and control the different steps of intracellular membrane traffic. More recently, they have been shown to also regulate cell signaling, division, survival, and migration. The regulation of these processes generally occurs through recruitment of effectors and regulatory proteins, which control the association of Rab proteins to membranes and their activation state. Alterations in Rab proteins and their effectors are associated with multiple human diseases, including neurodegeneration, cancer, and infections. This review provides an overview of how the dysregulation of Rab-mediated functions and membrane trafficking contributes to these disorders. Understanding the altered dynamics of Rabs and intracellular transport defects might thus shed new light on potential therapeutic strategies.

## 1. Introduction

Intracellular membrane trafficking is essential for the transport of membranes and cargoes between the different compartments in eukaryotic cells. It uses vesicular or tubular carriers that travel along the endocytic and exocytic pathways and it is regulated by complex protein machineries [1]. Rab GTPases are evolutionarily conserved regulators of vesicular transport, with more than 60 members described in humans [2,3]. They are localized to different membrane compartments in order to control both the specificity and the directionality of membrane trafficking. In this way, they confer membrane identity and ensure that cargoes are transported to their correct location within the cell [2,3]. Rab proteins exert their function by recruiting effector molecules such as coat proteins, motor proteins, kinases, phosphatases, and tethering factors to trigger downstream membrane trafficking events [4,5,6,7]. Similar to other small GTPases, Rab activity is tightly regulated by the cycling between an active GTP-bound and an inactive GDP-bound form [8]. The switch between these two states is mediated by regulatory proteins such as guanine nucleotide exchange factors (GEFs) that are responsible for the exchange of GDP for GTP, and GTPase-activating proteins (GAPs) that stimulate the Rab intrinsic GTPase activity. The GTPase cycle also allows Rabs to switch between being membrane-associated or cytosolic. Indeed, even though Rab proteins are prenylated at their C-termini for anchoring to membranes, the binding to a GDP dissociation inhibitor (GDI) keep them soluble in the cytosol. The recruitment of Rab proteins to membranes requires Rab activation by a GEF [9].

By regulating all the essential steps in membrane trafficking, Rab proteins are crucial not only for the maintenance of the correct cell homeostasis but also for specialized cellular functions. For example, in neurons, the regulation of intracellular trafficking is required for the functional synapsis and for the transport along neurites [10,11]. Indeed, different Rabs regulate the continuous flow of membranes at the synapsis, where synaptic vesicles undergo repeated cycles of exocytosis and endocytosis [12,13], and they also control the transport along axons and dendrites to and from the cell body for signaling and degradation [10,11]. 

Rab-mediated membrane transport has also a critical role in cancer, where alterations in the trafficking of integrins [14,15] or membrane type 1-matrix metalloproteinase (MT1-MMP) [16,17] result in the increase of the migratory and invasive ability of cancer cells. Intriguingly, Rab proteins additionally influence cell motility by directly modulating the cytoskeleton [6,18]. Furthermore, the role of Rabs in cancer also involves their emerging functions as regulators of cell division in both cytokinesis and abscission [19,20] and of signaling [21,22,23]. Thus the role of Rab proteins in cancer is not only related to defects in intracellular trafficking. 

In immune cells, Rab GTPases are involved in the regulation of the transport of immune receptors [24,25], secretion of chemokines and cytokines [26] and in the surveillance processes of endocytosis and phagocytosis [27,28] to modulate immune responses. 

Given the essential role of Rab proteins not only in membrane trafficking and cell homeostasis, but also in cell division and signaling, it is not surprising that alterations of these small GTPases are connected to a multitude of diseases. In addition to cancer [22,29,30,31], dysregulation or mutations in Rabs or their effectors has been associated with disorders such as neurodegerative diseases [10,11,32] and immune disorders [28]. Moreover, several pathogens have developed different strategies for intracellular survival that often involve the subversion of trafficking mechanisms altering Rab protein localization and function [33]. 

Recent reviews address the relevance of Rab proteins in one specific group of diseases (e.g., neurodegenerative disorders [10,11,32,34,35,36,37], cancer [29,38,39] or in immunity [28,33]), or the role of a specific Rab protein in these diseases [31,40,41,42,43]. Here, we provide a broader overview of the current knowledge on the role of Rab GTPases in multiple human diseases including neurodegenerative disorders, infections, and cancer, taking into account not only their well-established role as master regulators of membrane trafficking, but also their more emerging functions in cell signaling, migration, and cell division.

## 2. Rab Proteins in Neurodegenerative Diseases

The specialized morphology and function of neuronal cells is highly dependent on tight regulation of membrane trafficking. Indeed, in neurons, molecules and membranes need to be transported for long distances along dendrites and axons, as the latter can extend over a meter in length. Therefore, neurons have acquired specific mechanisms that regulate the transport of proteins, organelles, and receptors over long distances, in addition to the continuous cycles of exocytosis and endocytosis of synaptic vesicles at synapses [12,44]. 

Several Rab proteins are involved in the regulation of these processes in neurons. At the synapses, Rab5, Rab4, Rab11, and Rab35 regulate clathrin-mediated endocytosis for membrane retrieval, synaptic vesicle homeostasis, and recycling [45,46,47,48,49], while Rab3 and Rab27 regulate synaptic vesicle exocytosis and neurotransmitter release [50,51]. A few Rab GTPases also regulate the trafficking and turnover of neurotransmitter receptors at the post-synaptic site, where they control the delivery of neurotransmitters receptors (Rab8) [52], maturation of receptors (Rab39b) [53], insertion and removal at the plasma membrane (Rab4, Rab11) [54], and neurotransmitter surface abundance (Rab17) [55] (Figure 1). 

Rabs have important functions in both axonal anterograde and retrograde transport. Rab3, Rab27, and also Rab10 mediate the transport to the axon terminals, through the interaction with kinesin motor proteins [56,57,58]. In the opposite direction, Rab5 and Rab7a control the retrograde axonal transport of nerve growth factor (NGF), purinergic P2X3, and neurotrophin receptors [59,60]. Rab7a is also implicated in sorting of plasma membrane proteins for degradation at the axon terminal [61]. 

By regulating multiple transport pathways in neurons, Rab proteins are involved in dendrite branching and morphogenesis [62,63,64,65], neurite outgrowth [66,67,68,69,70], and neuronal migration during development [71,72]. As Rabs are fundamental for all these specialized functions in neurons, their misregulation is associated to several neurodegenerative diseases, including Alzheimer’s disease (AD), Parkinson’s disease (PD), amyotrophic lateral sclerosis (ALS), and Charcot–Marie–Tooth disease (CMT) [11,73] (Figure 1, Table 1).

### 2.1. Alzheimer’s Disease

Alzheimer’s disease (AD) is the most common form of neurodegeneration and dementia. Most AD cases are sporadic and not associated with genetic alterations or mutations, while 5–10% of AD cases are associated to mutations in Presenilin 1 (PSEN1), Presenilin 2 (PSEN2), and amyloid precursor protein (APP). These mutations lead to altered production of amyloid-β peptide (Aβ), which accumulates as extracellular deposits in the brain, and represent the main component of the Aβ plaques in AD pathogenesis, together with the neurofibrillary tangles [103]. 

PSEN1 and PSEN2 are proteases present in the endoplasmic reticulum (ER), trans-Golgi network (TGN), and vesicles, and are the catalytic component of the γ-secretase enzyme, which cleaves the APP into Aβ of varying lengths [104,105]. Mutations in either presenilin or APP consistently increase the relative ratio between the long and short Aβ amyloid peptides that assemble into neurotoxic oligomers [106,107]. PSEN1 is also involved in the regulation of membrane transport by binding to RabGDI, and in PSEN1-deficient neurons the RabGDI association with membranes decreases [108]. RabGDI binds to Rab proteins in the cytosol and deliver them to membranes, therefore a reduction in the amount of RabGDI associated with membranes would imply decreased levels of membrane-associated Rab GTPases. Indeed, in knockout cells for PSEN1 the levels of membrane-associated active Rab6 are decreased [74]. As Rab6 regulates the Golgi-to-ER transport pathway, it has been suggested that the reduced association of Rab6 to membranes could result in a defective recycling of vesicles from the Golgi to the ER in AD patients [74]. Evidence also exists for the role of Rab6 in the regulation of ER stress response in AD [75]. Intriguingly, AD mutations in presenilins increase susceptibility to ER stress [109], however the contribution of Rab6 to this function remains poorly characterized.

Defects associated to the functions of other Rabs are implicated in Aβ alterations, neurodegeneration and pathology in AD. For example, Rab11 and Rab3 were identified as key players in membrane-trafficking events regulating Aβ production, and a significant genetic association of Rab11 with late-onset AD suggests a causal link between Rab11 and AD [80]. Moreover, PC12 cells expressing the A260V mutant of PSEN1 are characterized by reduced expression levels of Rab8 and reduced Aβ production. In these cells, the lower production of Aβ is a consequence of the altered transport of the Aβ precursor, which accumulates in vesicles involved in TGN-to-PM transport [110]. This evidence highlights the effect of PSEN mutations or deficiency on Rab-mediated protein trafficking, which in turn can affect the production of Aβ.

On the other hand, sporadic AD cases are often associated with upregulation of Rab GTPase transcripts. For example, expression levels of early endosomal Rabs such as Rab4 and Rab5, the late endosomal Rab7a, and the exocytic Rab27, are upregulated in cholinergic basal forebrain neurons and correlated with cognitive decline in individuals with mild cognitive impairment and AD [77]. Rab5 and Rab7a overexpression was also reported in hippocampal CA1 neurons from individuals with AD [76]. The overexpression of Rab5 in these neurons is further associated with the downregulation of genes encoding the neurotrophin receptors TrkB and TrkC. This may lead to long-term deficits in hippocampal neurotrophic signaling, a key mechanism underlying AD that can be a possible new site for pharmacotherapeutic approaches [76,77]. In line with this hypothesis, it has been shown that the Rab5 effector APPL1 (adaptor protein containing pleckstrin homology domain, phosphotyrosine binding domain and leucine zipper motif) is involved in the overactivation of Rab5 in AD. This results in accelerated endocytosis, enlargement and impaired axonal transport of early endosomes, thus possibly interfering with the proper trafficking and signaling of neurotrophic factors [111]. Moreover, increased levels of Rab5 induce dysfunction of endosomes and are associated with an increase in APP cleavage and Aβ production, which are strongly implicated in AD [43,111]. These findings indicate that an overactivation of the endocytic pathway might lead to the defects in protein degradation and endosomal signaling associated with AD. 

Rab10 has also been linked to AD [40]. This small GTPase is involved in several intracellular trafficking pathways, including endocytic recycling, exocytosis, and early endosomal and post-Golgi transport, as well as in ER dynamics and morphology [112,113]. Rab10 is activated by phosphorylation at Tyr73 and this represents a pathological feature in the brains and dystrophic neurites of AD patients [78]. Even though the implications of this alteration are not known yet, silencing of Rab10 reduces Aβ, thus conferring protection against AD and representing a promising therapeutic target for AD prevention [79].

### 2.2. Parkinson’s Disease

Parkinson’s disease (PD) is characterized by the accumulation of intracellular inclusions known as Lewy bodies, containing aggregates of the protein α-synuclein (α-syn) and by neuronal loss in the substantia nigra of the brain, which causes striatal dopamine deficiency [114]. To date, several gene mutations have been associated to PD. Among these, a few are associated with α-syn, PTEN-induced putative kinase 1 (PINK1) and leucine-rich repeat kinase 2 (LRRK2), but also with Rab proteins [32,37]. One example is represented by loss-of-function mutations in the *Rab39b* gene [81,82]. Nonsense or missense mutations in Rab39b result in early onset PD [81,82,83,84]. Interestingly, Rab39b silencing in mouse hippocampal neurons reduces surface density of the α-amino-3-hydroxy-5-methyl-4-isoxazolepropionic acid (AMPA) receptor subunit GluA2 [53]. However, it has also been reported that Rab39b mutations are not a common cause of PD [115] and therefore the connection between Rab39b and PD needs further investigation.

Studies in animal models have contributed to elucidating the role of Rab proteins in PD. In the fruit fly *Drosophila melanogaster* carrying mutations in the *α-syn* gene, the overexpression of Rab11 is able to rescue the defects in vesicle size, neuronal loss, and motor impairments caused by the α-syn mutation, thus suggesting a therapeutic value of Rab11 in PD [116]. Indeed, Rab11 by interacting with α-syn, reduces its aggregation and toxicity. By regulating α-syn secretion and aggregation, Rab11 modulates two important mechanisms involved in PD [117,118].

Moreover, Rab8 also binds to α-syn and overexpression of either Rab8 or Rab1 is able to reverse α-syn-dependent impairment of ER–Golgi transport and PD-linked cytotoxicity in yeast, fruit flies, and *Caenorhabditis elegans* [119,120,121]. In line with this, silencing of Rab8, but also of Rab11, Rab13, or Rab39b, increases α-syn oligomerization whilst wild type and constitutively active mutants of the same Rabs decrease the number of cells with α-syn inclusions [117].

The involvement of Rab proteins in PD, besides their association with α-syn, is also connected to LRRK2. Little is known about the mechanisms affected by the mutations in the *LRRK2* gene in PD [122]. LRRK2 interacts with several RabGTPases including Rab7a, and it is involved in Rab7a-dependent endocytic trafficking and lysosomal positioning [87,88]. However, this function is impaired by the most common PD-causing LRRK2 mutation, thus linking endo-lysosomal dysfunction to the pathogenesis of LRRK2-mediated PD [87]. 

More recently, it has been discovered that LRRK2 also phosphorylates other Rab proteins, including Rab5, Rab3, Rab8, Rab10, Rab12, Rab35, and Rab43 [123,124], and dysregulation of Rab phosphorylation in the LRRK2 site causes neurodegeneration in primary neurons [125], thus suggesting a critical role for Rab GTPases and membrane trafficking in the LRRK2-caused PD. For example, pathogenic LRRK2 causes defects in the phosphorylation of Rab8a, resulting in an increased centrosomal localization of phosphorylated Rab8a, and deficits in centrosomal positioning and cohesion, with effects on neurite outgrowth, cell polarization, and directed migration [85]. LRRK2-mediated phosphorylation of Rab8a (and also Rab10) occurs upon Rab29 (also known as Rab7L1)-mediated activation of LRRK2, indicating that Rab29, by controlling LRRK2 recruitment to the Golgi and activation, is a key regulator of this PD predisposing kinase [89]. In line with this function, Rab7L1 is reported to be in a risk locus for sporadic PD, and higher expression of Rab7L1 is associated with higher risk of PD in humans [90,91].

Interestingly, Rab8a, Rab8b, and Rab13 are all phosphorylated in response to the activation of another key kinase, namely PINK1, whose mutations are causative of autosomal recessive PD [86]. Even though PINK1 does not directly phosphorylate these Rabs and the mechanisms underlying this pathway are poorly characterized, the phosphorylation of Rab8a and its isoform Rab8b downstream of PINK1 is abolished in patient-derived fibroblasts. This suggests that Rab phosphorylation upon PINK1 activation could be another mechanism involved in the neurodegenerative cascade of PD [86]. Furthermore, this indicates that both Rab8 isoforms are affected in Parkinson’s patients. 

However, Rab isoforms are not always implicated in the same disease. An example is Rab39, with Rab39b but not Rab39a involved in PD. Differently from Rab8a and Rab8b, Rab39a and Rab39b have distinct localization and control different processes within the cells [126]. Moreover, they have differential expression in tissues, with Rab39b highly expressed in brain tissue [126], thus indicating a critical role for this Rab isoform in the brain. This might therefore explain why loss-of-function mutations in the *Rab39b* gene, but not *Rab39a,* result in early-onset PD [81,82,83].

### 2.3. Amyotrophic Lateral Sclerosis

Amyotrophic lateral sclerosis (ALS) is a progressive neurodegenerative disorder, which specifically affects motor neurons in the motor cortex, brainstem and spinal cord. As membrane transport in motor neurons needs to cover long distances along axons, dysfunctions of the intracellular transport have critical effects and may contribute to the peculiar susceptibility of this disorder. Approximately 10% of ALS cases are familial and caused by mutations in specific genes. Among these, hexanucleotide repeat expansions (GGGGCC) in the first intron of the chromosome 9 open reading frame 72 (*C9orf72*) gene are the most common genetic cause of ALS, which is responsible for around 40% of familial ALS and 5–10% of sporadic ALS [127]. Haploinsufficiency for C9orf72 activity leads to motor neurons degeneration, thus implicating loss-of-function of C9orf72 in the ALS pathogenesis [128]. 

C9orf72 shows homology to the Differentially Expressed in Normal and Neoplasia (DENN) family of proteins, which are known Rab GEFs [129,130,131]. Indeed, C9orf72 has been reported to interact with Rab GTPases to regulate endosomal trafficking and also lysosomal biogenesis in motor neurons [128,132]. In motor neurons from ALS patients, C9orf72 is mainly localized to Rab5-positive early endosomes and the number of lysosomes is reduced, indicating a defect in endosomal maturation and lysosomal biogenesis [128]. Constitutively active Rab5 or chemical modulators of Rab5 effectors restore C9orf72 levels or increase its function, which in turn led to the recovery in patient neuron survival [128]. Also, constitutively active Rab5 ameliorates neurodegenerative processes in both gain- and loss-of-function C9orf72 mouse models [128], thus revealing Rab activity and lysosomal function as potential therapeutic targets for C9orf72-linked ALS. 

C9orf72 is a Rab1 effector with a role in controlling initiation of autophagy by regulating the Rab1-dependent trafficking of the autophagy initiation complex to the phagophore [92]. ALS patient-derived neurons present reduced levels of basal autophagy suggesting that defective autophagy could be one of the causes of ALS [92,133]. In line with this, depletion of C9orf72 impairs autophagy and leads to the accumulation of aggregates that are hallmarks of the ALS pathogenesis [133]. In addition, C9orf72 forms a complex with GEFs for Rab8 and Rab39b to control autophagic flux [133]. C9orf72 also interacts with Rab7a and Rab11, even though the function of these interactions is unknown [134]. 

Other ALS-causing mutations in *SOD1*, *TDP-43,* and *FUS* genes cause mis-localization of Rab GTPases including Rab1. Rab1 regulates the trafficking between ER and Golgi and is also involved in the unfolded protein response (UPR), thus the ALS-causing mutations in *SOD1*, *TDP-43* and *FUS* alter ER-Golgi transport and increase ER stress. Interestingly, overexpression of Rab1 exerts a protective effect against SOD1, TDP-43, and FUS mutation-caused ER-stress in ALS [93]. Hence, alteration of vesicle trafficking is a key factor in neurodegeneration caused by the *C9orf72* repeat expansion as well as *SOD1*, *TDP-43,* and *FUS* gene mutations in ALS pathology.

### 2.4. Charcot–Marie–Tooth 2B

Charcot-Marie-Tooth (CMT) disease is the most common hereditary peripheral neuropathy, which affects both motor and sensory nerves. More than 30 CMT disease-causative genes are known, and intriguingly several of them encode for proteins involved in intracellular traffic [135]. CMT2B is an axonal autosomal dominant form of CMT disease genetically linked to five missense mutations in the *Rab7a* gene [136,137,138,139]. The resulting mutants are characterized by faster nucleotide exchange rate and excessive activation [94,97,140]. As a consequence, CMT2B-causing Rab7a mutant proteins are mainly in the GTP-bound state and bind more strongly to effector proteins [97]. In contrast with these studies, work done on *D. melanogaster* suggested that CMT2B is a consequence of a partial loss-of-function of Rab7a, as expression of CMT2B mutants does not cause neuropathy-like phenotypes [141]. This raised the questions as to whether *Drosophila* is a good model for CMT2B or whether more optimal models, such as rodents and human neurons, are needed to more accurately define the mechanisms of the disease [142].

Rab7a is an ubiquitously expressed GTPase, present on late endosomes and lysosomes where it regulates late endocytic traffic, lysosome biogenesis, and autophagosome maturation [143]. Despite their high sequence similarity, Rab7a and Rab7b are not isoforms, as they have different intracellular localization and regulate different pathways [144,145]. In neurons, Rab7a is important for long-range retrograde axonal transport of neurotrophins and their receptors, and for neuritogenic signaling of the nerve growth factor (NGF) receptor TrkA [60,146]. Upon NGF binding, TrkA is internalized by endocytosis and retrogradely transported over long distances from the axonal synapse to the cell body and continues to signal, and Rab7a, by interacting with TrkA, controls both the trafficking and signaling of the receptor [146]. Endocytosis of the NGF–TrkA complex into signaling endosomes induces neurite outgrowth. Of note, CMT2B-causing Rab7a mutants promote premature degradation of TrkA and impair axonal trafficking of the receptor, resulting in the inhibition of neurite outgrowth [95,96], thus suggesting a possible mechanism which could contribute to CMT2B disease. In line with this, defective axonal transport has been reported in several model systems of CMT2B [147,148]. 

Other pathways that may contribute to CMT2B involve specific interactions of Rab7a with effectors selectively expressed in peripheral neurons, such as the intermediate filament protein peripherin. Peripherin is involved in neurite outgrowth and axonal regeneration after injury and CMT2B-causing Rab7a mutants bind more strongly to and increase the amount of soluble peripherin [149]. Therefore, the altered interaction of this intermediate filament protein with Rab7a mutants may influence the processes regulated by peripherin in CMT2B disease.

More recently, it has been shown that CMT2B-causing Rab7a mutants reduce autophagic flux, and that autophagy is inhibited in fibroblasts from a CMT2B patient, suggesting that alteration of the autophagic flux could be another mechanism responsible for neurodegeneration [98].

## 3. Rab Proteins in Cancer

Aberrant expression of Rab GTPases is often associated with cancer, especially for Rabs involved in endocytosis and recycling of adhesion molecules necessary for cell migration and metastasis, cell signaling, and cell division [29], (Figure 2, Table 2). 

Rab5 is overexpressed in highly proliferative and metastatic cancer cells and tissues. Its overexpression promotes cell proliferation and invasion by influencing focal adhesion kinase (FAK) in hepatocellular carcinoma and extracellular signal‑regulated kinase (ERK)/matrix-metallo proteinase-2 (MMP‑2) signaling pathways in oral cancer, respectively [150,151]. Besides Rab5, another early endosomal Rab, Rab21, also associates with α- and β- integrin chains thus regulating integrin-containing focal adhesions and therefore migration of cancer cells [153,170,171].

Another Rab involved in transport of integrins is Rab11. In particular, Rab11 mediates the trafficking of α6β4 integrin to the cell surface and hypoxia induces increased Rab11-dependent α6β4 integrin surface expression. Therefore Rab11 is thought to contribute to hypoxia-induced invasion of cancer cells [14]. Similarly, Rab5 and Rab22a are also involved in hypoxia-driven tumor cell migration, invasion, and metastasis, although the mechanisms involved are less characterized [152,162]. Interestingly, in neuronal axons, Rab11 and Rab coupling protein (RCP) control trafficking of β1 integrin to promote axonal extension, indicating that this mechanism is not unique to cancer cells [172].

Altered Rab-mediated vesicle trafficking can therefore cause enhanced cancer invasion. Another example is Rab8. Constitutively active Rab8 mediates the transport of exocytic vesicles carrying MT1-MMP to the plasma membrane for matrix degradation and invasion, while its knock-down prevents these processes [16]. Similarly, Rab2a, which is overexpressed in breast cancer, regulates the transport of MT1-MMP and E-cadherin, which promotes cancer spreading and invasion [17]. Rab40b is required for the secretion of two other metalloproteinase, MMP2 and MMP9, during invadopodia formation and for invadopodia-dependent extracellular matrix degradation [160]. Consistent with this, Rab40b upregulation correlates with the prognosis of gastric cancer by promoting migration, invasion, and metastasis [161].

Rab39a and its downstream effector Retinoid X Receptor Beta (RXRB), a member of the retinoid X receptor, are instead involved in cancer stemness regulation as silencing of Rab39a and inhibition of RXRB impairs tumorigenesis and cancer stemness [154]. In agreement with this, *Rab39a* is highly expressed in different types of tumors, including glioblastoma, glioma, lymphoma, leukemia, invasive breast cancers, and sarcomas, and genetic amplification of *RAB39A-RXRB* is often seen in various types of malignancies like breast and neuroendocrine prostate cancers [154].

Also altered expression of Rab23 has been reported in different types of cancers [163,164,165,166,167]. In more detail, its overexpression promotes cell migration and invasion through modulation of Rac1 activity, as well as cell proliferation [163,164,165,167]. Rab23 localizes to the plasma membrane and the endocytic pathway [173], however not much is known about its role in membrane traffic regulation. It is a negative regulator of Hedgehog signaling, and it seems to regulate the transport of essential components of this signaling pathway [169,173,174]. In further support of its role in signaling cascades, Rab23 is also involved in primary cilium transport [175,176], suggesting that its role in cancer could be related with signaling from the cilia. However, our actual knowledge on Rab23′s role as a negative regulator of hedgehog signaling is more in line with a function as a tumor suppressor. Even though Rab23 has been mainly reported to exhibit tumorigenic activity, evidence about a function as a tumor suppressor also exists [168]. Further investigation is needed to establish whether the role of Rab23 as an oncogenic protein or a tumor suppressor is cellular-dependent [177] and the molecular mechanisms behind these regulations. 

In addition to signaling pathways regulating cell migration and metastasis, Rab proteins also modulate signal transduction pathways that are implicated in survival and proliferation. An example is represented by Rab1a. Rab1a is overexpressed in colorectal cancer and hepatocellular carcinoma, and correlates with enhanced tumor progression, invasion and poor prognosis [21,155]. Rab1a is essential for the oncogenic growth by promoting mTORC1 signaling, a crucial node in signaling for cell survival, proliferation, and metabolism, in response to amino acid stimulation [155]. 

Recently, two somatic mutations in Rab35 have been identified in human tumors. Rab35 is an endocytic/recycling Rab and the identified Rab35 mutants, by regulating endocytosis and recycling, contribute to growth factor-mediated activation of phosphatidylinositol 3-kinase (PI3K) and protein kinase B (AKT) thus promoting cancer cell survival [22]. 

Depending on the types or subtypes of cancer, some Rabs have been described to be tumor promoter or tumor suppressor. An example is represented by Rab25, which enhances cell migration and invasiveness in ovarian and breast cancer cells possibly by controlling α5β1 integrin trafficking through recycling endosomes [15,156]. However, in colorectal carcinoma, triple-negative breast cancer, as well as in esophageal squamous cell carcinoma, Rab25 functions as a tumor suppressor and its loss promotes the development of intestinal neoplasia in mice [157,158,159]. One possible explanation for Rab25 cell type-dependent behavior could be the binding to cell type specific effector molecules. Indeed, another mechanism that influences cell migration and invasion in tumorigenesis involves the dysregulation of the interaction between Rab proteins and their effectors. For example, tumors with strong hypoxic signatures show low expression of a Rab5 effector, Rabaptin-5, which decelerates endocytosis due to the attenuation of Rab5-mediated early endosome fusion thus delaying EGFR signaling and contributing to oncogenesis [178].

## 4. Rab Proteins in Immune Diseases and Infections

Rab proteins have crucial importance for the correct function of cells of the immune system. Indeed, for a proper immune response, intracellular trafficking must be properly regulated in space and time, requiring the concerted efforts of several Rab GTPases [28,179]. Macrophages, dendritic cells, and neutrophils are critical players of the innate immune response. These cells engulf and destroy invading pathogens by phagocytosis and drive customized adaptive immune responses [180]. To respond to microbial infections, these cells need to recognize the pathogens. Upon binding to receptors localized on the surface of the cell, pathogens are internalized by phagocytosis. After pathogen uptake, the endolysosomal system contributes to the degradation of the cargo for antigen presentation, which is the basis of the adaptive immune response [181,182]. In addition to regulate the immune surveillance trafficking events by endocytosis and phagocytosis, Rab proteins are also important for the immune response since they are involved in the transport of immune receptors and the secretion of chemokines and cytokines [28]. Given their role in multiple immune-related trafficking events, alterations of Rab proteins are also implicated in the pathogenesis of infections and immune disorders. 

### 4.1. Infections

Many intracellular pathogens have evolved strategies to modulate the host intracellular trafficking pathways and escape degradation, and some of those targets are Rab proteins (Figure 3, Table 3). 

A well known example is *Mycobacterium tuberculosis (M. tuberculosis*), the causative agent of tuberculosis. After phagocytosis, *M. tuberculosis* is able to survive within the host cell by inhibiting the fusion of the phagosome with the lysosome [199]. More specifically, *M. tuberculosis* prevents the recruitment of the Rab5 effector early endosome antigen-1 (EEA1) blocking phagosomal maturation and the delivery of lysosomal enzymes needed for the degradation of the pathogen to the phagosome [179,183,200,201]. A secreted mycobacterial nucleoside diphosphate kinase acts as a GAP for Rab5 and Rab7a, thus preventing the recruitment of their respective effectors EEA1 and Rab-interacting lysosomal protein (RILP) to the phagosome, inhibiting phagosomal maturation and pathogen degradation [184].

The kinase LRRK2 has also an important role for in the immune system [202,203]. As mentioned above, LRRK2 is associated to PD and interacts with Rab proteins in neurons to regulate Rab7a-dependent endocytic trafficking and lysosomal positioning [87,88]. More recently, LRRK2 has been described to inhibit the maturation of *M. tuberculosis*-containing phagosomes in macrophages and impair innate immune responses [204], thus linking a Rab7a effector to both infections and neurodegenerative disorders. However, whether or not the underlying mechanisms in these different diseases are similar needs further investigation. 

Other Rab proteins are also recruited to the *Mycobacterium*-containing phagosome [205], however, their specific roles as well as the full mechanisms underlying the ability of mycobacteria to escape lysosomal degradation in the host cell are not fully characterized yet. 

Along the same line, the pathogen *Chlamydia trachomatis* enters host cells and establishes niche compartments named “inclusions”, which are required for its growth. To survive within these compartments, *C. trachomatis* secretes effector proteins into the host via a type III secretion system (T3SS) to modify the inclusion membrane. A protein named CT147, homologue of the Rab5 effector EEA1, tethers endosomes together but precludes their fusion because it lacks the Rab5 binding domain present on EEA1, thus blocking normal Rab5 recruitment, endosome fusion to lysosomes and bacteria degradation [185]. Other Rabs required for successful infection by *Chlamydia* are Rab11 and Rab14. These small GTPases are recruited by the adaptor protein Fip2 to the inclusion, even though the role of these interactions is unknown [206]. 

Rab38 and its homolog Rab32 have been reported to control *Salmonella* and *Listeria* infections [187,207,208]. These two Rabs coordinate the delivery of specific cargo to lysosome-related-organelles (LROs) including enzymes required for a variety of antimicrobial proteins. *Salmonella typhi* exhibits strict host specificity and can only infect humans. Intriguingly, the Salmonella-containing vacuole (SCV) exhibits features of LROs. It has been suggested that Rab32 and its GEF BLOC-3 may restrict *S. typhi* growth in mouse macrophages by delivering an antimicrobial activity to the SCV [187]. Broad-host *Salmonella enterica* serovars deliver two effectors, a Rab32 GAP and a protease specific for Rab29, Rab32 and Rab38 (the three Rabs present on the *S. typhi*-containing vacuole), to neutralize this pathway in mice, further supporting the importance of the Rab32/BLOC-3 pathway [186,188]. Differently from *Mycobacterium*-containing phagosome that avoids lysosomal fusion, SCV acquires many features of a lysosome, including acidic pH. However, SCV never acquires the mannose-6-phosphate receptor (MPR) because the *Salmonella* effector SifA sequesters Rab9 thus inhibiting the Rab9-mediated delivery of MPRs to the SCV and, consequently, also of the lysosomal enzymes transported by this receptor [189].

*Legionella pneumophila* is another intracellular bacterial pathogen that escapes degradation by establishing a replicative vacuole, the *Legionella*-containing vacuole (LCV). *Legionella* uses different effector proteins to modulates phagosomal trafficking and avoid fusion with lysosomes, including effector proteins that target Rab5, Rab22, and Rab1 [190,191,192,197]. In particular, *L. pneumophila* avoids fusion with lysosomes by manipulating host vesicular transport and recruiting Rab1 to LCV to create an ER-like compartment for bacterial replication [192,193,194,195]. In addition, *Legionella* effector proteins control Rab1 activity to ensure the proper spatial and temporal activation of this small GTPase [192,196,197,198].

### 4.2. Immune Disorders

Rab27a is associated with immune dysfunctions. Griscelli syndrome (GS) is a rare, autosomal recessive disorder, characterized by hypopigmentation of the skin and the hair, and accumulation of melanosomes in melanocytes. GS patients with mutations in *rab27a* gene also suffer from uncontrolled T-lymphocyte and macrophage activation. Furthermore, Rab27a-deficient T cells show reduced cytotoxicity and cytolytic granule exocytosis, which are essential pathways for immune homeostasis [209]. Defects in transport pathways regulated by Rab27a are also associated with another disease: choroideremia, an X-linked form of retinal degeneration. In particular, mutations in the Rab escort protein 1 (REP1), which is essential for prenylation of Rab GTPases, cause accumulation of unprenylated Rab27a [210]. 

Furthermore, defects in the transport pathways regulated by Rab13 seem to be involved in Chron’s disease (CD), a type of inflammatory bowel disease characterized by a chronic inflammation of the gastrointestinal tract. CD is characterized by alterations in cell-cell junctions resulting in loss of mucosal barrier integrity and increased intestine permeability, and the mistargeting of Rab13 to basolateral junctions in CD patients might be responsible of tight junction defects [211,212], (Table 4).

## 5. Conclusions

Rab GTPases are key regulators of membrane trafficking and cell homeostasis, and thus alterations of these proteins can lead to a multitude of diseases. It is indeed well established that aberrant expression, altered activity, or mistargeted localization of Rab proteins is associated with various disorders. 

Interestingly, alterations of the same Rab are present in different diseases. For example, defective Rab5 is found in both neurodegenerative disorders [76,77], cancer [150,151,152] and infections [183,184]. In AD, Rab5 is overactivated, leading to TrkB downregulation [76], while in cancer its overexpression influences either FAK or ERK/MMP2 pathways [150,151]. Furthermore, Rab5 activity can be modulated by pathogens to prevent phagosomal maturation and lysosomal degradation [184,185,190], thus indicating that defects in the same Rab can alter different pathways leading to different diseases. Similarly, alterations of Rab11 are associated with neurodegenerative disorders [99,100,101,102] and cancer [14]. This implies that some Rabs, such as Rab5 and Rab11, but also Rab4 and Rab7a, that regulate key pathways like endocytosis, vesicle recycling, and lysosomal degradation, have a major influence on cellular functions and when altered lead to various diseases. This is not surprising, as endocytosis has a fundamental role for synaptic functions [45,47,220], cancer signaling [221], and also pathogen internalization during infections [222]. 

Moreover, alterations of distinct Rabs can be implicated in the same disease. For example, several Rabs, including Rab7a, Rab5, Rab10, and Rab11, are associated to AD [76,77], indicating that defects in different Rab-mediated pathways can contribute to the same disease.

However, some disorders are characterized by disfunctions of a single Rab protein. The ubiquitously expressed Rab7a is the only Rab directly implicated in CMT, where its mutations affect only sensory and/or motor neurons [135]. This suggests that the CMT-associated Rab7a mutations mainly influence specific functions of these cells.

Mysteriously, often the malfunction of only a single Rab isoform is associated with a specific disease. An example is represented by Rab39, with Rab39a implicated in cancer [41] and Rab39b in PD [81,82,83,84]. An hypothesis is that the different expression of Rab isoforms in different tissues contributes to specific diseases, and in line with this the PD-associated Rab39b is highly expressed in the brain [126].

However, in most cases the molecular mechanisms underlying the pathogenesis are unclear, pointing to the necessity of further research. Therefore, the study of the basic mechanisms regulated by Rab GTPases is important to provide novel insights into the physiological roles of Rab proteins, but also for the proper understanding of pathogenic mechanisms. The increasing knowledge about the multiple functions of Rab GTPases, not only in the regulation of intracellular trafficking, but also in cell signaling, cell migration, and division, is dramatically improving the comprehension of the cell behavior and will hopefully contribute to the identification of new potential targets for therapeutic intervention. 

## Figures and Tables

**Figure 1 cells-08-00909-f001:**
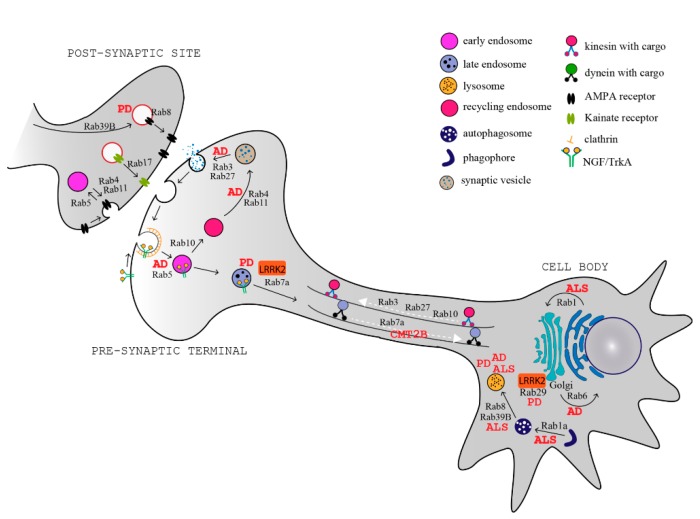
Rab proteins and transport pathways involved in neurodegenerative diseases. Schematic representation of a neuron and a post-synaptic terminal showing key Rab proteins and the trafficking pathways that they regulate. The Rab-dependent processes that are altered in neurodegenerative diseases (indicated in red) are highlighted. AD: Alzheimer’s disease, PD: Parkinson’s disease; ALS: amyotrophic lateral sclerosis; CMT2B: Charcot–Marie–Tooth 2B; LRRK2: leucine-rich repeat kinase 2; NGF: nerve growth factor; TrkA: tropomyosin receptor kinase A; AMPA: α-amino-3-hydroxy-5-methyl-4-isoxazolepropionic acid.

**Figure 2 cells-08-00909-f002:**
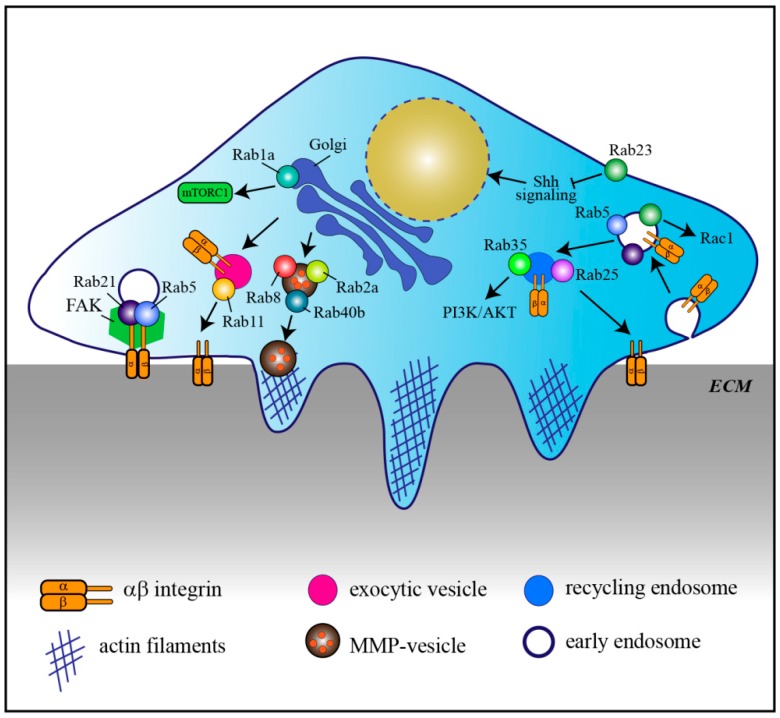
Rab proteins in cancer. Model representing a cancer cell invading the extracellular matrix (ECM) by using invadopodia, which are actin-rich structures. The illustration shows the main transport and signaling pathways affected by altered Rab protein expression in cancer. MMP: matrix metalloproteinase, FAK: focal adhesion kinase, Shh: Sonic hedgehog; PI3K: phosphatidylinositol 3-kinase; AKT: protein kinase B; mTORC1: mammalian target of rapamycin complex 1.

**Figure 3 cells-08-00909-f003:**
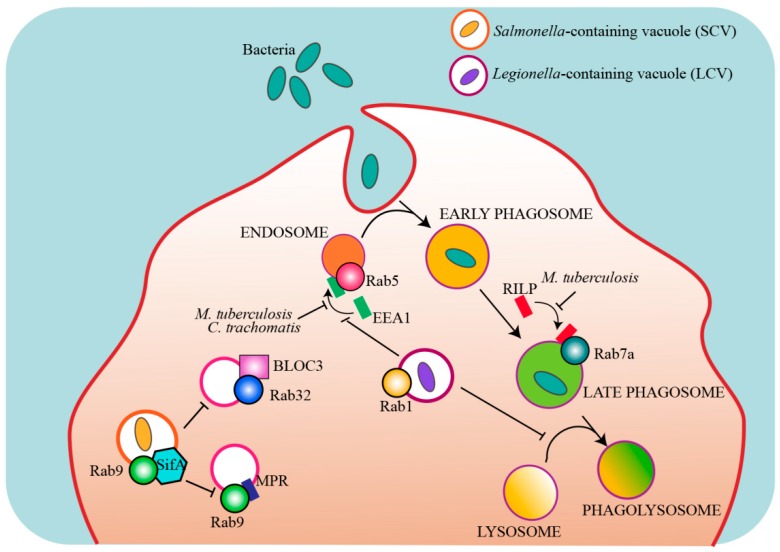
Rab proteins that are targeted by intracellular bacterial pathogens. Model showing the main Rab proteins and pathways that are targeted by intracellular bacterial effectors to escape lysosomal degradation and establish a replicative organelle. EEA1: early endosome antigen-1, RILP: Rab-interacting lysosomal protein, BLOC-3: biogenesis of lysosome-related organelles complex-3, SifA: Salmonella-induced filament-A, MPR: mannose 6-phosphate receptor.

**Table 1 cells-08-00909-t001:** Rab proteins involved in neurodegenerative diseases.

	Rab Protein	Rab Alterations Associated with the Disease	References
Alzheimer’s Disease (AD)	Rab6	Increased levels but decrease of active, membrane-associated Rab6 and defective recycling of vesicle	[74,75]
	Rab5	Upregulated and over-activated	[76,77]
	Rab7a	Upregulated	[76,77]
	Rab4	Upregulated	[77]
	Rab27	Upregulated	[77]
	Rab10	Phosphorylated and upregulated	[78,79]
	Rab11	Genetic association of Rab11 with late-onset AD; affects Aβ production	[80]
Parkinson’s Disease (PD)	Rab39b	Loss-of-function mutations	[81,82,83,84]
	Rab8a/b	Defective phosphorylation by pathogenic leucine-rich repeat kinase 2 (LRRK2) leading to centrosomal defects and influencing neurite outgrowth, cell polarization and migration. Abolished phosphorylation in PTEN-induced putative kinase (PINK1)-caused PD	[85,86]
	Rab7a	Decreased activity; defective endosomal trafficking and degradation in LRRK2-caused PD	[87,88]
	Rab29 (Rab7L1)	Upregulated and present in a risk locus for sporadic PD; over activates LRRK2 PD-mutants by increasing their recruitment to the Golgi	[89,90,91]
Amyotrophic Lateral Sclerosis (ALS)	Rab1	Dysfunctional in sporadic ALS; its effector C9orf72 is mutated in ALS, resulting in a decreased autophagy. Rab1-dependent endoplasmic reticulum (ER)-Golgi transport inhibited in superoxide dismutase 1 (SOD1), TAR DNA binding protein (TDP-43), and Fused in Sarcoma (FUS)-associated ALS.	[92,93]
Charchot–Marie–Tooth 2B (CMT2B)	Rab7a	Missense mutations characterized by excessive activation that cause a reduced autophagic flux, premature neurotrophine receptor degradation, impaired axonal trafficking of the receptor, and inhibition of neurite outgrowth	[94,95,96,97,98]
Huntington’s Disease (HD)	Rab11	Decreased activity and defects in endosomal recycling	[99,100,101,102]

**Table 2 cells-08-00909-t002:** Rab proteins involved in cancer.

Rab Protein	Rab Alterations Associated with Cancer	References
Rab5	Overexpressed, promotes invasion by influencing FAK signaling and the extracellular signal-regulated kinase (ERK)/matrix metalloproteinase 2 (MMP2) pathway; required for hypoxia-driven tumor cell migration, invasion, and metastasis	[150,151,152,153]
Rab11	Hypoxia promotes Rab11-mediated trafficking of α6β4 integrin to the cell surface contributing to tumor cell invasion	[14]
Rab39a	Overexpressed; genetic amplification of *RAB39A-RXRB*	[154]
Rab1a	Overexpressed; promotes oncogenic growth by activating mammalian target of rapamycin complex 1 (mTORC1) signaling	[21,155]
Rab35	Somatic mutations; regulates endocytosis and recycling, contributing to phosphatidylinositol 3-kinase (PI3K) and protein kinase B (AKT) activation and promoting cell survival	[22]
Rab25	Altered expression: overexpression enhances cell migration and invasiveness by regulating α5β1 integrin trafficking. Downregulation possibly influences multiple pathways	[15,156,157,158,159]
Rab40b	Overexpressed, mediates secretion of MMP2/9 thus promoting cancer invasion and metastasis	[160,161]
Rab2a	Overexpressed; regulates the transport of membrane type 1 metalloproteinase (MT1-MMP) and E-cadherin and promotes invasion	[17]
Rab22a	Overexpressed, involved in hypoxia-driven tumor migration, invasion and metastasis	[162]
Rab23	Altered expression: overexpression induces cell migration, invasion and proliferation by modulating Rac1 activity. Downregulation possibly affects the hedgehog signaling pathway	[163,164,165,166,167,168,169]

**Table 3 cells-08-00909-t003:** Intracellular bacteria that target Rab proteins.

	Targeted Rab Protein	Mechanism	References
*Mycobacterium tuberculosis*	Rab5	Secretes a nucleoside diphosphate kinase that acts as GAP for Rab5, preventing early endosome antigen 1 (EEA1) recruitment. This arrests phagosome maturation	[183,184]
	Rab7a	Secretes a nucleoside diphosphate kinase that acts as GAP for Rab7a, preventing Rab-interacting lysosomal protein (RILP) recruitment. This inhibits lysosomal enzyme transport and phagosomal degradation	[184]
*Chlamydia trachomatis*	Rab5	Secretes CT147, homologue of EEA1, that prevents endosomal fusion and bacterial degradation	[185]
Broad-host *Salmonella enterica*	Rab32	Delivers a Rab32 GAP and a Rab32-specific protease to neutralize the Rab32/ biogenesis of lysosome-related organelles complex-3 (BLOC-3) antimicrobial pathway	[186,187]
	Rab29	Delivers a Rab29-specific protease	[188]
	Rab38	Delivers a Rab38-specific protease	[187]
	Rab9	Delivers SifA, an effector that sequesters Rab9, inhibiting the delivery of mannose 6-phosphate receptors (MPRs) and lysosomal enzymes to the Salmonella-containing vacuole (SCV)	[189]
*Legionella pneumophila*	Rab5	Secretes a GEF for Rab5 and the effector VipD that prevents the binding of Rab5 to downstream effectors blocking endosomal trafficking and lysosomal degradation	[190,191]
	Rab21	Secretes a GEF for Rab21	[190]
	Rab22	Secretes a GEF for Rab22 and VipD that prevents the binding of Rab22 to downstream effectors blocking endosomal trafficking and lysosomal degradation	[190,191]
	Rab1	Secretes GEF for Rab1 to recruits Rab1 to SCV and effectors that regulate Rab1 cycle to create a replicative organelle	[192,193,194,195,196,197,198]

**Table 4 cells-08-00909-t004:** Rabs implicated in immune disorders and other diseases.

	Rab Protein	Alterations	Reference
Griscelli syndrome	Rab27a	Mutations cause melanosome accumulation in melanocytes and uncontrolled T-lymphocyte and macrophage activation	[209]
Choroideremia	Rab27a	Accumulation of unprenylated Rab27a causes defects in transport pathways regulated by Rab27a	[210]
Crohn’s disease	Rab13	Dislocated to the basolateral junctions in the intestinal epithelium, possibly affecting intestinal permeability	[212]
Carpenter’s syndrome	Rab23	Truncating and missense mutations	[213,214]
Cone-Rod dystrophy	Rab28	Nonsense mutations cause phagocytosis defects in cones	[215]
X-linked mental retardation	Rab39b	Loss-of-function mutations lead to defective neurite growth cones and reduced presynaptic buttons	[126]
Warburg Micro syndrome	Rab18	Loss-of-function mutations in Rab18 or its GEFs	[216,217,218,219]

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
