# Peer review of "Rab GTPases: Switching to Human Diseases"

_cells, 2019, doi:10.3390/cells8080909_

Round 1

Reviewer 1 Report

Guadagno et al.,on

Rab GTPases: switching to human diseases

Rab proteins compose the largest family of small GTPases and control the different steps of intracellular membrane traffic. Alterations in Rab proteins and their effectors are associated with multiple human diseases. In this review, the authors aimed to provide an overview of how the dysregulation of Rab contributes to these disorders.

While I believe that this is a very interest area to be covered by a review, many similar reviews have been published recently. To be useful, this review has to distinct itself from the recently published ones. For example, several reviews have been published in 2018-2019 regarding Rabs and Alzheimer’s disease/ Parkinson’s disease. This review did not yield any new information or insight beyond previous ones. As such, I strongly suggest the author to focus this review on areas not covered extensively by other recently published reviews, such as Cancers.

Specific concerns:

In the abstract, the author stated that: More recently, they(rabs) have been shown to regulate also cell signaling, division, survival and migration. It is interesting to know if these other effects are due to the disrupted intracellular trafficking or independent the role of Rabs in intracellular trafficking. I would suggest the author to discuss this in their introduction section.

At the end of the introduction section, I suggest the author to point out and cited those recently published reviews regarding Rabs and diseases (There are more than 20 in last two years), and stated how this review is different from others and worth to be read.

While the authors have described various reports regarding the involvement of Rab in various diseases, I would like to see a comprehensive analyze and/or summarize of these findings in a Summary and/or Conclusion section (current conclusion section is lack of substances). For example, as a reader, I would like to know if Rabs’ role in various disease are similar or different, if there are several Rabs that are more important/relevant to diseases than the others, If different Rabs are involved in different diseases.

Author Response

Rab proteins compose the largest family of small GTPases and control the different steps of intracellular membrane traffic. Alterations in Rab proteins and their effectors are associated with multiple human diseases. In this review, the authors aimed to provide an overview of how the dysregulation of Rab contributes to these disorders.

While I believe that this is a very interest area to be covered by a review, many similar reviews have been published recently. To be useful, this review has to distinct itself from the recently published ones. For example, several reviews have been published in 2018-2019 regarding Rabs and Alzheimer’s disease/ Parkinson’s disease. This review did not yield any new information or insight beyond previous ones. As such, I strongly suggest the author to focus this review on areas not covered extensively by other recently published reviews, such as Cancers.

      We thank the referee for the observation. However, we have followed the editor’s suggestion for the choice of the review’s topic. We agree that several reviews about Rab proteins have been published in the last years. Nevertheless, most of them are either focused on the role of Rab proteins in specific diseases (e.g. in Alzheimer’s or Parkinson’s disease as pointed by the referee) or on the role of a single Rab protein in diseases. In this review, we have tried to present a broader and more comprehensive overview about the role of Rab proteins in several diseases including neurodegenerative diseases, cancer and immune diseases/infections. We have now clarified this, as suggested, in the introduction (lines 63-65).

Specific concerns:

In the abstract, the author stated that: More recently, they(rabs) have been shown to regulate also cell signaling, division, survival and migration. It is interesting to know if these other effects are due to the disrupted intracellular trafficking or independent the role of Rabs in intracellular trafficking. I would suggest the author to discuss this in their introduction section.

Following the referee’s suggestion, we have now discussed this point in the introduction (lines 46-52)

At the end of the introduction section, I suggest the author to point out and cited those recently published reviews regarding Rabs and diseases (There are more than 20 in last two years), and stated how this review is different from others and worth to be read.

As suggested, we have now clarified how this review is different from others. There are indeed several recent reviews regarding Rab and diseases but they are mainly focused on the role of one specific Rab in diseases or on the role of Rabs on a specific disease, as now pointed in the introduction (lines 63-65).

While the authors have described various reports regarding the involvement of Rab in various diseases, I would like to see a comprehensive analyze and/or summarize of these findings in a Summary and/or Conclusion section (current conclusion section is lack of substances). For example, as a reader, I would like to know if Rabs’ role in various disease are similar or different, if there are several Rabs that are more important/relevant to diseases than the others, If different Rabs are involved in different diseases.

We thank the referee for the excellent suggestion. We have now expanded the Conclusion section as indicated (lines 499-526).

Reviewer 2 Report

Comments for Authors

Guadagno and Prodiga have written a comprehensive review of the role that Rab GTPases play in human disease. It is well written with clear and well-designed figures. I recommend that it be accepted with the following minor revisions.

The authors should refer to the recently identified role for LRRK2 as a negative regulator of M. tuberculosis phagosome maturation (EMBO J. 2018 Jun 15;37(12)) in section 4.1.

The following grammatical errors should be corrected;

a.       Line 24: Rab GTPases are evolutionarily conserved regulators of……..

b.       Line 48: …it is not

c.       Line 168: ‘…associated with PD.’

d.       Line 198 – 201: The sentence beginning ‘Rab8a (and Rab10)…..’ is confusing. Rephrase.

e.       Line 334: ‘however not…..’

f.        Line 345: ‘modulate signal transduction pathways that are implicated in….’

g.       Line 347: ‘poor prognosis’

h.       Line 421: ‘by inhibiting…’

i.         Line 467: ‘also suffer from….’

Author Response

Guadagno and Prodiga have written a comprehensive review of the role that Rab GTPases play in human disease. It is well written with clear and well-designed figures. I recommend that it be accepted with the following minor revisions.

The authors should refer to the recently identified role for LRRK2 as a negative regulator of M. tuberculosis phagosome maturation (EMBO J. 2018 Jun 15;37(12)) in section 4.1.

We thank the referee for the positive comments. We have now referred to the function of LRRK2 in the regulation of M. tuberculosis phagosome maturation (lines 436-442).

The following grammatical errors should be corrected;

Line 24: Rab GTPases are evolutionarily conserved regulators of…….. Line 48: …it is not Line 168: ‘…associated with PD.’ Line 198 – 201: The sentence beginning ‘Rab8a (and Rab10)…..’ is confusing. Rephrase. Line 334: ‘however not…..’ Line 345: ‘modulate signal transduction pathways that are implicated in….’ Line 347: ‘poor prognosis’ Line 421: ‘by inhibiting…’ Line 467: ‘also suffer from….’

We have now corrected all the indicated grammatical errors as well as others.

Reviewer 3 Report

In this manuscript, Guadagno and Progida provided a comprehensive and updated review about roles of Rab GTPases in health and disease. The authors succeeded to conjoin basic knowledge of Rab-mediated trafficking pathways with recent progress that has been made in connecting Rab disorders with a multitude of diseases, such as neurodegenerative diseases, infections and cancer. The review is well structured, carefully researched, understandingly written and nicely illustrated. Given that the mechanisms of Rab dysregulation in human diseases are not definitely disclosed, the review also challenges continuative investigations. 

Specific Comments:

In my view, the manuscript would benefit if the authors will more clearly address the issue of Rab isoforms. Most of the 60 Rab members in humans are known to exist in at least two isoforms. Accordingly, it appears mysterious why malfunctions of certain Rab isoforms are associated with diseases (e.g. Rab27A and Charcot-Marie-Tooth disease) without functional complementation by their structurally and functionally related paralogs (e.g. Rab27B).

Minor comments.

Fig. 1. Please explain the abbreviation LRRK2 in the figure legend.

Fig. 2. Please explain the abbreviation mTORC1 in the figure legend.

Fig. 3. What is meant: Phagolysosome instead of Phagolysome?

There are a few typos:

Line 68: a verb is missing.

Line 327: Why is Rab39a in italic?

Line 329: Why is Rab39A-RXRB in italic?

Line 334: not instead of mot.

Line 349: mode instead of node?

Line 421: inhibiting instead of inhabiting.

Author Response

In this manuscript, Guadagno and Progida provided a comprehensive and updated review about roles of Rab GTPases in health and disease. The authors succeeded to conjoin basic knowledge of Rab-mediated trafficking pathways with recent progress that has been made in connecting Rab disorders with a multitude of diseases, such as neurodegenerative diseases, infections and cancer. The review is well structured, carefully researched, understandingly written and nicely illustrated. Given that the mechanisms of Rab dysregulation in human diseases are not definitely disclosed, the review also challenges continuative investigations.

Specific Comments:

In my view, the manuscript would benefit if the authors will more clearly address the issue of Rab isoforms. Most of the 60 Rab members in humans are known to exist in at least two isoforms. Accordingly, it appears mysterious why malfunctions of certain Rab isoforms are associated with diseases (e.g. Rab27A and Charcot-Marie-Tooth disease) without functional complementation by their structurally and functionally related paralogs (e.g. Rab27B).

We thank the referee for the useful suggestion. We have now addressed the issue of Rab isoforms in both section 2.2 (lines 218-223), and  in the discussion (lines 522-526). In addition, we have specified that Rab7a and Rab7b despite their names are not isoforms (lines 277-279), thus explaining why CMT2B is characterized by mutations only in Rab7a and not Rab7b.

Minor comments.

Fig. 1. Please explain the abbreviation LRRK2 in the figure legend.

Fig. 2. Please explain the abbreviation mTORC1 in the figure legend.

Fig. 3. What is meant: Phagolysosome instead of Phagolysome?

There are a few typos:

Line 68: a verb is missing.

Line 327: Why is Rab39a in italic?   

We have used the italic font when we refer to gene names, like in this case

Line 329: Why is Rab39A-RXRB in italic?

As before, we have indicated gene names in italic

Line 334: not instead of mot.

Line 349: mode instead of node?

We actually meant “node”, but we have now rephrased the sentence to make it clearer.

Line 421: inhibiting instead of inhabiting.

We have now corrected all typos as well as explained the indicated abbreviations. 

Round 2

Reviewer 1 Report

The authors have adequately revised MS to address my major concerns.